# Systemic Barriers to Optimal Cancer Care in Resource-Limited Countries: Jordanian Healthcare as an Example

**DOI:** 10.3390/cancers16061117

**Published:** 2024-03-11

**Authors:** Razan Mansour, Hikmat Abdel-Razeq, Maysa Al-Hussaini, Omar Shamieh, Akram Al-Ibraheem, Amal Al-Omari, Asem H. Mansour

**Affiliations:** 1Department of Internal Medicine, University of Kansas Medical Center, Kansas City, KS 66103, USA; rmansour2@kumc.edu; 2Department of Internal Medicine, King Hussein Cancer Center (KHCC), Amman 11941, Jordan; habdelrazeq@khcc.jo; 3Department of Pathology, King Hussein Cancer Center (KHCC), Amman 11941, Jordan; mhussaini@khcc.jo; 4Department of Palliative Care, King Hussein Cancer Center (KHCC), Amman 11941, Jordan; oshamieh@khcc.jo; 5Department of Nuclear Medicine and PET/CT, King Hussein Cancer Center (KHCC), Amman 11941, Jordan; aibraheem@khcc.jo; 6Department of Scientific Affairs and Research, King Hussein Cancer Center (KHCC), Amman 11941, Jordan; asomari@khcc.jo; 7Department of Diagnostic Radiology, King Hussein Cancer Center (KHCC), Amman 11941, Jordan

**Keywords:** cancer care, Jordan, LMIC, cancer disparities, refugees, barriers

## Abstract

**Simple Summary:**

In the face of significant hurdles stemming from conflicts and resource constraints, Jordan has emerged as an exemplary paradigm, solidifying cancer care both locally and regionally. Despite the positive progress being made, a significant obstacle is emerging in the form of increasing rates of cancer, which is representative of a larger upsurge in non-communicable diseases. However, a thorough grasp of the various contributing factors is still lacking. Therefore, we aimed to carefully examine and discuss the current identified barriers that are hindering the achievement of optimal cancer care in Jordan.

**Abstract:**

This narrative review explores the multifaceted barriers hindering access to quality cancer care in Jordan. A literature-based narrative review was undertaken to explore the current identified barriers to cancer care in Jordan. Four databases were searched using relevant keywords to identify key insights on barriers and proposed solutions. Key challenges and potential solutions were identified based on evidence from studies, reports, and initiatives. Medical services and infrastructure exhibit centralized disparities, impacting rural and underserved areas. Human resources shortages, geopolitical instability, and quality management issues pose significant challenges. Public awareness campaigns face hurdles in addressing the tobacco epidemic and late-stage diagnosis. Socioeconomic disparities, particularly in health insurance and urban–rural divides, further compound barriers. Refugees encounter distinct challenges, including late-stage diagnosis, financial barriers, and psychological distress. Despite multiple challenges, Jordan presents a model for regional development and health equity. This study not only contributes to improving cancer care in Jordan but also offers a roadmap for policymakers, healthcare practitioners, and researchers in similar contexts globally. Government initiatives, financial aspects, and proposed policy measures are examined as potential solutions. Recommendations include coordinated prevention strategies, enhanced screening uptake, training programs, the equitable distribution of facilities, and policy directives aligned with global commitments. The role of digital technologies, telemedicine, and community engagement models is emphasized.

## 1. Introduction

Jordan faces unique healthcare challenges as a low-middle-income country (LMIC) [1]. Located in the Eastern Mediterranean region with a land area of approximately 89,000 km, Jordan’s current population is estimated to be 10 million, with an annual population growth rate of 2.4% [2]. According to statistics from the Jordanian Cancer Registry, cancer is a leading cause of morbidity and mortality in Jordan, with breast and lung cancer being the most prevalent in women and men, respectively [3]. The cancer burden is compounded by high rates of major noncommunicable diseases (NCDs) like cardiovascular disease, diabetes, and hypertension. Demographic and economic pressures, such as poverty, urbanization, and the influx of refugees, strain Jordan’s healthcare system [1].

Healthcare comprises approximately 8% of gross domestic product (GDP) expenditure in Jordan [4]. Jordan’s healthcare is mainly provided through the public and private sectors. The public sector is composed of the Ministry of Health (MOH), the military’s Royal Medical Services, and university hospitals. The private sector accounts for approximately 33% of all hospital beds [5]. Specialized cancer care is centralized at the King Hussein Cancer Center (KHCC), which serves over 60% of cancer patients in Jordan [6]. Private insurance does not cover cancer treatment, and the government in Jordan assumes the costs of caring for Jordanian patients with cancer at public hospitals and the KHCC. However, barriers across the healthcare system, the economic landscape, and demographics impede equitable access to quality cancer care and put additional strain on patients with cancer seeking equitable and comprehensive cancer care.

This narrative review will establish the need to examine these barriers in the Jordanian context. It will preview key challenges within medical services, human resources, quality management, public awareness, and socioeconomics. The review will outline evidence-based solutions to overcome these barriers, guided by examples from the KHCC and global best practices, to ultimately enhance cancer prevention, early detection, and comprehensive treatment for the Jordanian population.

## 2. Materials and Methods

### 2.1. Data Sources and Validity

The data for this narrative review were gathered from a search of peer-reviewed academic literature, government reports, and international organization databases. Four public databases, namely PubMed, EMBASE, Scopus, and Web of Science, were searched using relevant keywords related to cancer care barriers in Jordan. Furthermore, the records of Google Scholar served as a supplementary resource to ensure a more comprehensive exploration of relevant studies. The search was manually conducted with the assistance of a medical librarian, and a detailed search strategy for databases can be found in the Appendix A. Additionally, non-peer-reviewed reports from organizations such as the World Health Organization (WHO), the Jordanian Ministry of Health, the KHCC, and the Jordanian Cancer Registry were retrieved. All data sources used in this review were available in the public domain and were deemed valid based on peer-review processes, government oversight, and institutional credibility.

### 2.2. Review Conduct

Studies were reviewed to assess their eligibility based on their relevance to the topic of cancer care barriers in Jordan. The authors included the studies with relevant information on barriers to cancer care, including medical services and infrastructure, human resources, quality management, public awareness, socioeconomic disparities, and the specific challenges faced by vulnerable populations, with a focus on refugees. The data review included information on the nature of barriers, contributing factors, and proposed solutions. The review covered a period spanning from inception to 3 January 2023, providing a comprehensive analysis of the barriers to cancer care in Jordan over the past two decades.

### 2.3. Analysis Approach

Data analysis involved synthesizing information from diverse sources to identify common themes, patterns, and key insights related to barriers to cancer care in Jordan. Thematic analysis techniques were employed to categorize and organize findings according to thematic areas, including healthcare infrastructure, human resources, quality management, public awareness, socioeconomic disparities, and refugee healthcare. The review also examined proposed policy measures and initiatives aimed at overcoming identified barriers, highlighting successful strategies and areas requiring further attention.

## 3. Results

### 3.1. Medical Services and Infrastructure

The healthcare system in Jordan is divided between the private and public sectors, with 41 public and 65 private hospitals. However, the centralized and disconnected nature of cancer care services poses major barriers in Jordan. Both diagnostic and treatment facilities are concentrated in major cities, creating disparities for rural and underserved governorates [7]. Fragmented care across institutions and poor connectivity for sharing medical records hinder timely diagnosis and coordinated treatment [8]. Electronic medical records have been previously proposed as a solution to enable streamlined referrals and information exchange, but implementation challenges remain, mainly due to lack of funding and resources [7]. Structural and behavioral obstacles also lead to pronounced rural–urban disparities, including poorer health literacy on cancer symptoms and limited engagement with preventative services [9]. Initiatives to expand rural facilities and address provider absenteeism are needed [10], as suboptimal referral systems further delay cancer care as patients struggle to navigate between public, private, and non-governmental facilities [11].

### 3.2. Human Resources

Jordan faces shortages of specialized oncology professionals and trained oncology nurses, which strains existing staff and compromises patient care quality, and the associated rise in nurse–patient ratios impacts patients’ experiences and outcomes [12]. Due to this shortage, patients in rural areas may face challenges in accessing specialized cancer treatment due to the uneven distribution of healthcare professionals [10]. The shortage of oncologists and oncology nurses results in longer waiting times for cancer diagnosis and treatment, leading to delays in care and poorer outcomes [13]. Moreover, the emigration of oncologists and cancer specialists also cripples local capacity while draining global resources [7]. This brain drain is driven by limited career development and local incentives. Outdated training programs also contribute to suboptimal skills with newer technologies among healthcare workers. The KHCC faces staffing challenges due to the migration of highly qualified healthcare workers, mainly younger physicians seeking better opportunities abroad [14]. Despite efforts to retain talent, the allure of higher incomes and training opportunities in Western countries has led to a significant brain drain. This trend reflects global disparities in healthcare resources between LMICs like Jordan and high-income countries (HICs) [14].

### 3.3. Quality Management

Geopolitical instability in the region poses sustainability challenges for Jordan’s cancer care system. Political and economic fluctuations compromise consistent quality by disrupting supplies, healthcare funding, and service delivery [7]. Several barriers, including resource constraints, infrastructure deficiencies, regulatory gaps, and limited patient engagement, pose significant challenges to the implementation of quality management initiatives [12]. There are also disparities in care quality between private and public institutions, partly due to varying accreditation and standardization [15]. Centralized reporting and quality benchmarking are lacking in Jordan, which hinders the delivery of more uniform patient care. Robust primary care is also essential for effective cancer prevention and management but remains underdeveloped in Jordan [7]. Primary services that are equally distributed play a key preventive role in promoting healthy lifestyles and early detection [16]. Suboptimal reporting also impedes quality improvement efforts, as there are limitations in data accuracy and standardized quality metrics reported in Jordan [17].

### 3.4. Public Awareness

Public awareness of cancer is essential for promoting early detection, timely treatment, and better outcomes [18]. Misconceptions and misinformation about cancer may prevail, leading to delays in seeking medical attention and reluctance to engage in preventive behaviors [19]. The tobacco epidemic remains a major risk factor for cancer in Jordan, with over 60% of males smoking [20]. The high prevalence of smoking is associated with a high incidence of lung cancer, which is the leading cause of cancer death in Jordan. Besides lung cancer, tobacco use also increases the risk of cancers of the head and neck, colon, bladder, and acute leukemia; all are among the most common neoplasms in Jordan [21,22]. Despite public health efforts, smoking rates remain high and further expansion of smoke-free policies, advertising restrictions, and youth-focused interventions are needed [21]. Limited access to reliable sources of information, including healthcare providers, educational materials, and media campaigns, hinders efforts to raise awareness and promote early detection [23,24,25]. Late-stage diagnosis also impedes survival, driven by low rates of cancer screening, limited health literacy on symptoms, and healthcare access barriers (Figure 1) [7,22].

Support services for cancer survivors are also not well integrated, including home-based care, psychosocial support, and rehabilitation [7].

### 3.5. Socioeconomic Disparities

#### 3.5.1. Health Insurance Disparities

There are significant gaps in health insurance coverage in Jordan driven by socioeconomic status. Only about 65% of Jordanians have some form of health insurance, with lower rates among unemployed, rural, and refugee populations [26]. Informal and agricultural workers lack access to government or private insurance schemes, resulting in high out-of-pocket health expenditures [27]. These disparities may lead to treatment delays, incomplete courses of therapy, or the abandonment of treatment altogether, compromising patient outcomes [26]. Uninsured patients face greater financial barriers to treatment, contributing to poorer survival rates compared to insured patients [28]. Government initiatives expanding insurance to schoolchildren and subsidizing premiums have reduced but not eliminated health insurance disparities, especially across geographic regions [29].

#### 3.5.2. Urban–Rural Disparities

There are pronounced disparities in healthcare access between urban and rural areas in Jordan, with around 20% of the Jordanian population living in rural areas [16]. In Jordan, data from the World Bank showed a drop in the rural population from 49% in 1960 to 16% in 2015, constituting 8.8% of the total population in 2019 [30]. Rural populations face geographic barriers to reaching cancer diagnostic and treatment facilities that are concentrated in major cities [31]. Long distances and transport costs contribute to lower utilization of services and higher physician turnover [10]. Health literacy is also lower in rural communities, and illiteracy rates are higher than the national rate (6.7%) [2]. This is likely attributed to poor awareness of cancer symptoms and limited engagement in preventative behaviors [32,33]. High rates of provider absenteeism further exacerbate access challenges in rural regions [34]. Factors like poor work environments and lack of supervision contribute to absenteeism among rural health workers, which directly reduces service availability and quality [10]. These factors drive patients living in rural areas to seek services in the urban areas of Jordan; however, the lack of robust public transportation and the poor road network between the rural and urban areas of the county hinder the interaction between the rural and urban areas of Jordan [35].

### 3.6. Refugee Population and Cancer Care

Jordan has the largest population of refugees per capita in the world, with Syrians comprising the majority of the refugee population in Jordan [36]. Refugees face multiple challenges when it comes to cancer care in Jordan [37]. The Syrian refugee crisis has deeply affected the surrounding region since its onset in 2011, affecting more than five million Syrians that have sought refuge outside their homeland, predominantly in Turkey, Jordan, and Lebanon [38]. Studies from Jordan have shown that refugees often present at more advanced stages and receive suboptimal treatment [39,40]. Factors include disruptions in prior treatment, cost barriers, social marginalization, and discrimination [41]. Additionally, some refugees face structural barriers to receive care, such as not possessing the appropriate civil documentation for treatment [42]. Refugee care also relies heavily on humanitarian funding from United Nations agencies and international nongovernmental organizations, which remains below needs [43]. Despite government subsidies, refugees in Jordan have reported unaffordable out-of-pocket costs and long waiting times for care [44]. Refugees outside camps face particular disadvantages in accessing healthcare and navigating the system [45], and initiatives to expand insurance coverage, provide transport assistance, and strengthen primary care services for refugees are lacking in the region.

#### 3.6.1. Late-Stage Diagnosis and Treatment Challenges

Cancer treatment in refugees involves a number of challenges, including financial stress, lack of health records, and frequent relocating and moving around [44]. Low awareness of cancer warning signs or symptoms, and stigma about cancer, persist in certain refugee populations in Jordan [19,46]. These issues make access to expensive oncological surgery, radiation, and chemotherapy difficult if not impossible for patients with long therapy plans, significantly complicating the provision of quality care [19]. In a study by Abdel-Razeq et al. on Syrian refugees seeking cancer treatment in Jordan, most patients with breast cancer had locally advanced or metastatic disease on presentation. More than one third of the patients had deviations from KHCC treatment guidelines and had worse outcomes than the general Jordanian population (Figure 2) [39].

#### 3.6.2. Access to Timely and Optimal Treatment

It is of note that refugees also tend to have a higher incidence of delayed and suboptimal cancer treatment. One study found that the timing in receiving treatment varies from as low as 25% up to as high as 85%, with suboptimal timing of treatment in about 60% of patients [47]. These percentages illustrate only one aspect of this multidimensional problem; in addition to the aforementioned systemic barriers, refugees’ lack of capacity to navigate the system of a new country on their own makes expensive treatments like chemo-radio-surgery impossible without help or aid provided from outside sources [19]. Refugees also face challenges with timely access to treatment due to the grave cost of medical care and out-of-pocket medical costs stemming from clinical workups, multiple treatments, admission with complications, and indirect expenses such as accommodation and transportation [36]. A cost analysis in 2017 described the expenses linked to cancer treatment for Syrian refugees living in host nations using a per capita cost method, with an estimated total cost of 140.23 million euros in 2017 [48].

The medical services available within refugee camps managed by governmental bodies and non-governmental organizations are often relatively basic, lacking specialized providers and equipment to diagnose cancers or provide multimodal therapy, with suspected cases requiring referral to tertiary hospitals outside the camps after initial presentation and evaluation [49]. However, some recent global health initiatives have focused on facilitating direct oncology health services for refugees inside camps in Jordan through partnerships between humanitarian groups like Médecins sans Frontières (MSF) and national cancer treatment centers, to enable a more streamlined care.

#### 3.6.3. Psychosocial and Survivorship Support

Refugees as a population report extremely high rates of adverse mental health conditions like major depression, post-traumatic stress disorder (PTSD), anxiety, and adjustment disorders that ultimately stem from prior violence exposure, traumatic loss of home/assets, family separation, uncertainty about the future, culture shock, and discrimination [50,51]. Most cancer care facilities in Jordan lack structured programs for psychosocial support [52]. This significant psychological distress and impaired quality of life amplifies the emotional burden typically associated with a concurrent cancer diagnosis. Additional culture-specific grief like losing one’s land also requires a nuanced understanding. Unfortunately, survivorship programs tailored specifically to promote rehabilitation and coping with cancer among refugee groups are markedly lacking, as seen from an assessment of the current global health literature, and are sorely needed to help address refugees’ unique psychosocial needs and facilitate community reintegration [53,54].

### 3.7. Government Initiatives and Financial Aspects

#### 3.7.1. Government Subsidies and Investment

The Jordanian government has undertaken various initiatives to subsidize cancer treatment, recognizing the financial burden it imposes on patients [52]. Cancer treatment is offered at no cost to all Jordanian citizens through public hospitals including the MOH, Royal Medical Services, university hospitals, and the KHCC [52]. This approach is in line with global efforts to improve accessibility to lifesaving healthcare services. Notably, the government has been investing in increasing its health infrastructure, especially in cancer care and cancer screening. For example, one governmental initiative provided mobile mammography services to rural areas in Jordan that provided free-of-cost breast cancer screening to people who were eligible for breast cancer screening [55]. Such initiatives stand as an example of the commitment to a total service in cancer care, instead of a “quick fix” to direct cancer treatment. According to Marzouk et al. (2019), the uneven distribution of healthcare facilities is of paramount importance in equalizing access to health services [44]. However, the study concluded that it could be challenging to access underserved areas, and constant monitoring is important for the elimination of such disparities. A study carried out by Abdel-Razeq et al. highlighted the difficulties faced in the timely provision of cancer care among Syrian refugees, particularly by advocating for equal regional expansion drives [39]. Progress in this area has been made; however, more efforts have to be exerted so that these gaps can be closed and the whole population can benefit from better cancer care services.

#### 3.7.2. Health Insurance Coverage

Health insurance coverage plays a crucial role in facilitating access to cancer care services. In Jordan, about 65% of Jordanians have some form of health insurance, with lower rates observed among unemployed individuals and rural populations [26]. There is coverage through the Health Insurance Fund, the Military Health Insurance Fund, and private medical insurance. Studies that explored the impact of health insurance status on cancer outcomes in Jordan revealed that uninsured cancer patients faced greater financial burdens, leading to delays in appropriate cancer screening or seeking treatment compared to insured patients [56,57]. Lack of health insurance coverage is associated with higher rates of advanced-stage cancer at diagnosis and increased mortality rates among cancer patients [39]. Prior studies in the literature have emphasized the importance of addressing gaps in health insurance coverage to improve cancer care delivery, as well as the need for policy interventions aimed at expanding health insurance coverage to vulnerable and underserved populations, such as rural communities and refugees, to ensure equitable access to cancer treatment services [58,59].

#### 3.7.3. Financial Assistance Programs

Financial assistance programs play a crucial role in ensuring access to cancer care services for individuals in Jordan. However, disparities and limitations in these programs can hinder equitable access to treatment and support for cancer patients. Al-Qadi and Lozi examined the effectiveness of financial assistance programs for cancer patients in Jordan [60]. Their study highlighted challenges such as limited funding, eligibility criteria, and bureaucratic processes, which hinder access to financial support for many patients. Additionally, Hammad et. al. explored the impact of financial barriers on cancer care access in Jordan. The study identified gaps in existing financial assistance programs, particularly in terms of coverage for treatment costs, transportation, and supportive care services [17]. Another study investigated the outcomes of financial assistance programs provided by the Goodwill Cancer Care Fund in Jordan. Their findings indicated that while the fund has helped alleviate financial burdens for many cancer patients, there are still significant gaps in coverage and accessibility, particularly for marginalized populations [61].

### 3.8. Efforts to Overcome Barriers

In 2018, KHCC mobile clinics visited the remote and poor parts of Jordan to provide initial free detection services for the most common forms of cancer. Mobile clinics indeed have been highly effective, screening more than 5000 patients per year in communities plagued by poverty and with shortages of facilities for cancer care [26]. A study showed that over 75% of refugee patients came to the KHCC with a late-stage cancer diagnosis [40]. KHCC has strategically taken initiatives towards early confirmation through awareness of warning signals and free screening programs offered. It helps lower delays in diagnosis and initiates timely treatment, and this practice has been proven to be effective in enhancing outcomes of cancer prevention. Additionally, KHCC has been very proactive in launching campaigns in mass media across Jordan, which revolve around cancer awareness and promoting early detection. These campaigns, through television, radio, and social media or via community outreach programs, have targeted the whole population, not just the high-risk segments. Studies of awareness campaigns have indicated improved knowledge and attitudes related to cancer care [52]. Various specific educational programs focus on the stigmatization of cancer, which impedes patients from accessing proper care in a timely manner. KHCC has also linked up with religious leaders, teachers, and support groups to help reinforce the message that cancer can be treated if detected early. Since then, social awareness campaigns have done much in terms of de-stigmatizing cancer to encourage patients to receive screening services [62].

Since 1997, the Goodwill Fund set up by KHCC has been of help in distributing more than 150 million dollars’ worth of treatment costs to over 25,000 financially impoverished individuals [26]. By covering cancer treatment costs, it has helped eliminate financial barriers that can impact clinical outcomes. Despite the significant impact, the rising cancer burden in Jordan and the region has made it difficult for charitable organizations to adequately meet the increasing demand for financial assistance. Effective resource utilization and partnerships with government agencies can help expand coverage. A recent agreement signed between KHCC and Jordan’s Ministry of Health in 2022 has allowed the expansion of cancer care to disadvantaged regions through a new comprehensive cancer center to the east of Amman (Aqaba), operated by KHCC and starting in 2024. The new cancer facility will provide chemotherapy, palliative care, and pathology services to cancer patients in the eastern regions of Amman that lack specialized cancer care centers. This is a major milestone in addressing disparities by facilitating access to quality cancer care. Local non-governmental organizations like the Jordan Breast Cancer Program have played an instrumental role in tackling barriers through free early detection services, financing treatment costs, and implementing awareness campaigns targeted at low-income groups. Their field presence and ties to local communities have helped overcome disparities. Non-governmental organizations have advocated and developed partnerships with stakeholders, including KHCC and government agencies, to implement national policies focused on cancer prevention and early detection, especially among disadvantaged groups. Their collaborative efforts have been crucial for impacting cancer care on a larger scale in Jordan.

## 4. Discussion

This study holds significance as the first study in the region to provide a comprehensive understanding of barriers to cancer care across medical services, healthcare infrastructure, workforce, quality management systems, public awareness, and socioeconomic disparities (Table 1).

There is an urgent need for coordinated, integrated prevention strategies encompassing lifestyle changes, vaccination against oncogenic infections, and evidence-based preventive measures tailored to the local context.

As the first of its kind focused on Jordan, the findings of our review serve as a foundation for addressing gaps in cancer care within the country. Moreover, the study’s insights are highly applicable to LMICs facing similar healthcare challenges and can be extrapolated to the Middle East and North Africa (MENA) region. While each country may have its unique challenges, several common themes emerge across the region. In the MENA region, limited health insurance coverage poses a significant barrier to accessing cancer care. Studies in LMICs have shown that a lack of comprehensive health insurance schemes results in high out-of-pocket expenses for cancer treatment, leading to financial burdens that deter patients from seeking timely care [63,64,65]. This problem is exacerbated by the absence of government-funded healthcare programs tailored to cancer prevention, screening, diagnosis, and treatment. For example, in Egypt, where health insurance coverage is limited and out-of-pocket payments are common, access to essential cancer services remains a challenge for many individuals, particularly those from low-income backgrounds [66].

The inadequacy of healthcare infrastructure is another critical barrier to cancer care in LMICs. Limited access to oncology centers, diagnostic facilities, and essential medical equipment hinders early detection and timely treatment initiation [67,68,69]. In Iraq, the healthcare infrastructure has been severely affected by years of conflict and underinvestment, resulting in limited access to specialized cancer care services, particularly in rural areas [70]. The shortage of trained healthcare professionals, including oncologists, nurses, and allied healthcare workers, poses a significant challenge to cancer care provision in LMICs [67,71,72]. In countries like Syria and Yemen, ongoing conflicts and political instability have disrupted medical education and training programs, leading to a severe scarcity of qualified personnel capable of delivering comprehensive cancer care services [73]. Disparities in access to cancer care between urban and rural areas are pervasive in LMICs. In Saudi Arabia, there is a significant disparity in cancer care access between urban and rural areas, with rural populations facing challenges related to limited healthcare infrastructure and transportation barriers [74,75]. However, efforts to improve cancer care access in rural areas have been hampered by logistical challenges and disparities in healthcare resource allocation [76].

Governmental policies and regulatory frameworks play a crucial role in shaping cancer care delivery in LMICs. However, weak governance, corruption, and a lack of political will to prioritize cancer control initiatives hinder progress in this area [77,78]. Jordan has implemented national cancer control strategies and policies to improve cancer care delivery and outcomes, despite facing challenges related to limited resources and refugee healthcare [52]. The provision of cancer care to refugees presents unique challenges in LMICs hosting large refugee populations. In Turkey, which hosts millions of Syrian refugees, efforts to provide cancer care to refugees have been hindered by language barriers, legal restrictions, and limited access to healthcare services in refugee camps [79,80]. Similarly, in Lebanon, where a large refugee population resides, integrating refugee healthcare into national cancer control programs has been challenging due to resource constraints and legal barriers [81].

With these challenges to cancer care in LMICs comes the need for proposed solutions. Primary healthcare facilities can play a pivotal role through public education, promoting cancer screening, and early detection services. Concerted efforts are required to enhance cancer screening through cost subsidies, mobile clinics, and awareness campaigns [82]. Bridging public awareness gaps will involve coordinated action across policymakers, healthcare institutions, and grassroots organizations. Cancer prevention and control campaigns should encompass risk factor mitigation, early detection promotion, and holistic survivorship support.

Targeted outreach programs are vital to increasing screening rates among underprivileged communities where late-stage diagnosis remains high. Initiatives to improve rural postings through rotations, infrastructure upgrades, and incentive schemes could help address absenteeism. In addition, targeted enrollment campaigns, premium subsidies, and engagement of civil society groups could help extend coverage to disadvantaged communities [83,84]. Eliminating disparities will require ongoing monitoring and equitable resource allocation across all population groups. Tackling rural–urban cancer disparities requires improving geographic accessibility, community engagement, and health workforce capacity. Investments in outreach services, patient transport networks, telehealth solutions, and provider training and support are warranted. Partnerships between urban centers and rural facilities can also help transfer knowledge and resources to underserved areas. A balanced approach across all regions is key for equitable cancer outcomes.

Specialized training programs and incentives are crucial to address shortages of oncologists and cancer care professionals, especially in disadvantaged regions. Refining referral pathways through centralized coordination and enhanced communication between institutions could minimize treatment delays and improve outcomes [8]. Partnerships between academic institutions and healthcare facilities can help build localized capacity through structured residency and fellowship programs. Improved retention policies, including competitive compensation, research opportunities, and global collaborations, could help address this challenge [14]. There needs to be an equitable distribution of human resources between urban and rural locales enabled through needs-based healthcare planning and infrastructure expansion [85]. This can help eliminate disparities in access to quality cancer care based on geographic location. National policies need to align with global commitments toward equitable access to affordable, quality cancer services spanning prevention, treatment, palliation, and survivorship support. Policy directives can help eliminate financial and social barriers to cancer care. Policies must integrate evidence-based survivorship care models encompassing psychosocial support, management of late effects, screening for recurrence, and wellness education. This can help improve health outcomes and quality of life after cancer treatment.

Refugees face additional impediments to treatment and cancer care in LMICs, including late-stage diagnosis, increased costs, and suboptimal medical and mental support services. Proposed services should combine medical, mental health, complementary medicine, community/religious engagement, and refugee cancer patient peer support aspects to provide appropriately holistic support following primary treatment that cancer centers in high-income nations may take for granted. In addition, to address the barrier of lacking structured programs for psychosocial support in cancer care facilities, we propose the establishment of comprehensive psychosocial oncology programs that aim to provide emotional and spiritual support to both patients and their families, enhancing their quality of life and ensuring optimal treatment outcomes. Dedicated social workers, collaborating closely with healthcare management teams, can effectively address the emotional and social needs of patients and their families, thereby facilitating access to treatment and improving treatment compliance.

Digital technologies such as telemedicine, mHealth applications, and connected medical devices can help make access to quality cancer care services easier, especially for patients living in remote areas. Tele-oncology models might improve results by enabling remote consultations, online tumor boards, and remote monitoring in patients undergoing chemotherapy. Scaling such models, though, requires cross-sector coordination and the ability to develop digital infrastructure with an urgent need to design and implement national guidelines for screening common cancers that should be differentiated according to the local disease burden and resource context [86]. This can significantly enhance early detection. Access to free public education programs aimed at making the community aware of common cancer symptoms and benefits associated with early illness detection can also more significantly contribute because they will help enhance screening adoption rates in order to downstage cancer diagnosis. The vision should move from a short-term, disease-based care model to multidisciplinary patient-centered care throughout the cancer continuum that involves health promotion, prevention, end-of-life care, and survivorship based on individual needs [87]. Grassroots-based community engagement models are crucial as they enable local leaders or advocates for health to gain the capacities related to campaigning on reasons around cancer prevention and control and can accelerate progress in capacity building. Comprehensive monitoring and evaluation systems are also critical for identifying gaps in care quality management and outcomes. The adoption of unified performance indicators could better guide quality enhancement initiatives across Jordan’s cancer care system. We also propose to have simplified metrics for the objective evaluation of the quality of care so that the results can be shared to identify gaps and acquire best practices.

Advocating for broadening the scope of financial assistance is of paramount importance. Prior studies have highlighted that the problem of financial obstacles to various diseases has raised the need for comprehensive policies related to public healthcare needs [17,60]. A strategic recommendation would be to improve partnerships between government and non-governmental organizations such as the KHCC and the Jordanian Breast Cancer Program, which could help and support government programs. These partnerships can provide cancer patients with comprehensive financial support for cancer care [88]. It is also necessary to monitor and assess financial aid programs continuously in order to identify their effectiveness, and to improve financial assistance mechanisms to keep them updated with the evolving demands from the population and health system [89,90].

The comprehensive cancer care infrastructure in Jordan in the 1990s was almost non-existent, with services scattered throughout Amman, and there were no initiatives for prevention or early detection [52]. This resulted in late diagnosis and adverse outcomes. The year 1997 marked the creation of the KHCC, which encouraged radical changes in cancer care by setting up a full-fledged cancer facility that provided equitable and affordable access to quality diagnosis and treatment as well as palliative care corresponding with worldwide standards [62]. KHCC has been an innovative platform to elevate national principles of cancer treatment in Jordan as the standard-setting center. KHCC’s patient-centered approach, interdisciplinary model of care delivery, and comprehensiveness of services, along with its dedication to innovation, have provided care to thousands of patients over the last two decades. For instance, it led critical countrywide programs like the Jordan Breast Cancer Program in 2003 that substantially contributed towards breast cancer downstaging via diagnosis and improved results [91]. In 2006, KHCC received international accreditation by Joint Commission International (JCI) as a symbol of global safety and quality; in 2019 KHCC earned Magnet accreditation, which has been linked to superior clinical results; and in 2020 KHCC received AAHRPP accreditation for quality research and its protection of participants.

KHCC is on a continuous upgrade path, constantly adding the relevant tech to its arsenal, like intraoperative MRI neurosurgery rooms, numerous linear accelerators, and imaging equipment such as positron emission tomography/computed tomography (PET/CT) for a complete services package [92,93]. Stereotactic radiosurgery has been one of the most advanced techniques that emerged in the field of radiation oncology. KHCC has also been recognized as a training center for top-flight companies such as Elekta and Philips in new radiotherapy technologies. According to the latest Jordan Cancer Registry report, 10,000 new cancer cases were registered at KHCC in 2019, of which 75.9% were Jordanian citizens, representing substantial national reach [2]. Over 20% of cancer patients came from neighboring countries, reflecting KHCC’s emergence as a reputable tertiary care destination in the region despite local and regional challenges [94]. Managing an international population presents unique challenges but also allows experience-sharing with global centers.

Drawing upon lessons learned from Jordan, a sustainable cancer care system in LMICs can be designed to address key challenges while maximizing available opportunities (Table 2).

A comprehensive healthcare delivery model would integrate primary care services with specialized cancer centers to ensure seamless patient management and coordinated multidisciplinary care [95]. Equitable access can be achieved by strategically allocating healthcare facilities and resources based on population needs, leveraging mobile clinics and telemedicine to bridge geographical gaps and improve access [49,94].

Capacity building involves investing in training and retaining a skilled healthcare workforce through collaborative partnerships with international institutions and tailored educational programs [9,12]. Some LMICs have established cancer registries at the federal level; however, the quality control mechanisms of these registries are extremely variable, and quite often data is either missing or lags behind current data entry mechanisms and long term outcomes, which leads to considerable delays in updating the data and would preclude accurate mortality or morbidity analysis. In addition, unless the burden of each cancer type and subtype is known, it is hard to allocate the appropriate resources for the prevention or treatment of the specific types of cancer. The need for accurate data of cancer cases in a country is imperative to develop cancer control programs and screening guidelines [96,97].

Financial support strategies include strengthening existing government initiatives and non-governmental partnerships to expand financial assistance programs, ensuring inclusivity in health insurance coverage and providing targeted support for vulnerable populations [26,44]. In addition, prioritizing prevention and early detection entails evidence-based public education campaigns and subsidized screening services [82,98]. Quality improvement initiatives would establish robust monitoring and benchmarking systems to track care quality across healthcare institutions, implementing standardized performance metrics and best practices informed by international standards and research evidence [11,17].

Medical training and research efforts would foster a culture of research infrastructure and collaborative partnerships between academia, healthcare institutions, and industry stakeholders [14,85]. Community engagement strategies would involve engaging local communities and grassroots organizations in cancer prevention and control efforts through culturally sensitive interventions and community-based participatory research approaches [45,46]. Additionally, ensuring comprehensive palliative care services integrated within the cancer care continuum is crucial. This includes addressing the physical, psychosocial, and spiritual needs of patients and their families, providing symptom management, pain relief, and end-of-life care [99,100]. By implementing evidence-based strategies within this framework, Jordan can overcome existing challenges and optimize its cancer care system to ensure equitable access to high-quality care for all residents, while acknowledging and addressing resource constraints and limitations.

This review possesses several strengths that enhance its significance and credibility, as it represents the first comprehensive review focused specifically on Jordan, providing a unique perspective on the country’s cancer care barriers. Moreover, its findings hold relevance beyond Jordan, serving as a valuable resource for other resource-limited LMICs confronting similar healthcare challenges, particularly in the MENA region. However, the study is not without its limitations. Potential publication bias may exist due to the reliance on the available literature, and the exclusion of non-English studies may limit the generalizability of the findings. The focus on barriers alone may overlook potential facilitators, while contextual differences between countries may restrict the direct applicability of lessons learned from Jordan to other countries. Additionally, the reliance on secondary data sources may constrain the depth of analysis and the ability to capture real-time changes in the healthcare landscape.

## 5. Conclusions

Key barriers to cancer care include shortages of specialists, maldistribution of facilities, financial constraints, low cancer awareness, and socioeconomic factors. To build equitable cancer care, strategies like investments, policy changes, and partnerships between the public and private sectors are needed. Research should focus on expanding specialized human resources and creating healthcare facilities. National cancer control plans should use technology to initiate awareness campaigns. Lessons learned from Jordan can inform strategies for improving cancer care access, quality, and outcomes in other thriving LMICs.

## Figures and Tables

**Figure 1 cancers-16-01117-f001:**
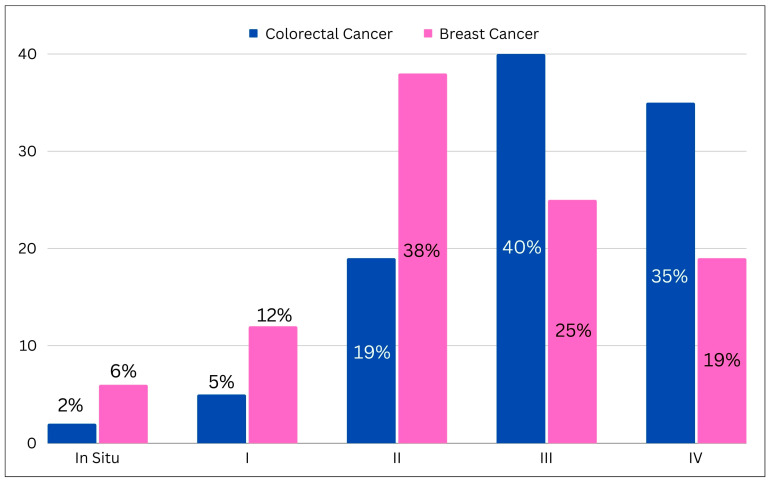
Bar chart illustrating the initial stage of colorectal and breast cancer from the King Hussein Cancer Center registry (I: Stage 1; II: Stage 2; III: Stage 3; IV: Stage 4).

**Figure 2 cancers-16-01117-f002:**
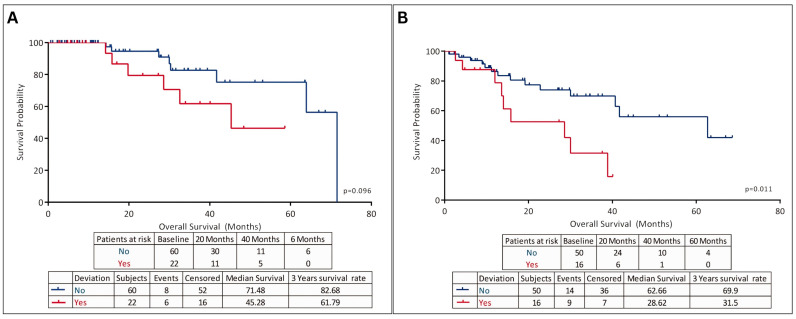
Analysis of (**A**) overall survival, and (**B**) disease-free survival in Syrian refugees with breast cancer who sought treatment at KHCC (adapted from [39]).

**Table 1 cancers-16-01117-t001:** Summary of main barriers to cancer care in Jordan.

Barriers	Description
Medical Services and Infrastructure	Concentration of diagnostic and treatment facilities in major cities, causing disparities in rural areas. Fragmented care across institutions with poor connectivity for sharing medical records. Lack of electronic medical records hindering timely diagnosis and coordinated treatment. Pronounced rural–urban disparities in health literacy and engagement with preventative services. Suboptimal referral systems.
Human Resources	Shortages of specialized oncology professionals and trained oncology nurses. Migration of qualified healthcare workers abroad for medical training. Brain drain due to limited career development and local incentives. Outdated training programs leading to suboptimal skills with newer technologies. Lack of continuous education in the rapidly changing landscape of cancer care.
Quality Management	Geopolitical instability affecting the consistency in quality of cancer care. Disparities in care quality between private and public institutions. Underdeveloped primary care. Suboptimal reporting, data accuracy, and standardized quality metrics. Lack of comprehensive monitoring and evaluation systems.
Public Awareness	High rates of smoking contributing to the tobacco epidemic. Late-stage diagnosis driven by low screening uptake, limited health literacy, and healthcare access barriers. Limited access to reliable sources of healthcare providers, educational materials, and media campaigns. Misconceptions and misinformation lead to delays in seeking medical attention.
Socioeconomic Disparities	Significant gaps in health insurance coverage based on socioeconomic status. High out-of-pocket health expenditures for uninsured patients. Government initiatives reducing but not eliminating coverage gaps. Disparities often lead to treatment delays, incomplete courses of therapy, or abandonment of treatment.
Urban–Rural Disparities	Pronounced disparities in healthcare access between urban and rural areas. Geographic barriers, transport costs, and lower health literacy contributing to lower utilization of services in rural areas. Provider absenteeism in rural regions impacting service availability and quality.
Vulnerable Populations	Refugees facing barriers such as advanced stages at diagnosis, cost barriers, discrimination, and reliance on humanitarian funding. Disruptions in prior treatment, social marginalization, and discrimination. Lack of integrated support services for refugee cancer patients. Low cultural competence needed to address stigma and discriminatory attitudes.
Governmental and Financial Aspects	Lack of political will to prioritize cancer control initiatives. Healthcare facilities with uneven distribution. Absence of universal healthcare coverage. Financial assistance programs present to a select patient population. Need for continuous analysis of the scope and effectiveness of financial aid programs. Need for a robust national cancer registry. Limited partnerships between government and non-governmental organizations.

**Table 2 cancers-16-01117-t002:** Proposed solutions to address barriers to cancer care in low-middle-income countries (LMICs).

Barriers	Proposed Solutions	Responsible Entity
Medical Services and Infrastructure	Invest in primary healthcare facilities with cancer screening programs, diagnostic facilities, and treatment capabilities. Provide cost subsidies for cancer screening services and establish mobile clinics to reach remote areas. Upgrade existing healthcare infrastructure and establish evenly distributed facilities.	Ministry of Health, healthcare institutions, policymakers
Human Resources	Specialized training programs for oncologists, nurses, and allied healthcare professionals. Implement retention policies, including competitive salaries and personal and professional development opportunities. Collaborative partnerships and twinning programs between local and international healthcare institutions.	Ministry of Health, medical/nursing schools, healthcare institutions
Quality Management	Robust monitoring systems with standardized performance metrics. Implement a benchmarking process to compare care quality with national and international standards. Promote a culture of continuous improvement by identifying areas for enhancement and implementing best practices.	Ministry of Health, institutional leaders, quality improvement agencies
Public Awareness	Evidence-based public education campaigns to raise awareness about cancer prevention and early detection. Provide subsidies or free screening services to encourage individuals to undergo regular cancer screening tests. Engage community leaders and influencers to disseminate information about cancer prevention and control.	Non-governmental organizations, primary care providers, Ministry of Health, healthcare institutions
Socioeconomic Disparities	Expand financial assistance programs to provide support for cancer treatment costs. Strengthen health insurance coverage to ensure equitable access to cancer care services. Community-based support programs for individuals and families facing financial challenges due to cancer diagnosis and treatment.	Ministry of Health, health insurance agencies, non-governmental organizations
Urban–Rural Disparities	Mobile clinics and telemedicine solutions to bring cancer care services closer to rural communities. Outreach programs and community engagement activities to raise awareness about cancer and facilitate access to healthcare services in rural areas. Invest in improving healthcare infrastructure in rural areas.	Ministry of Health, local health departments, policymakers
Vulnerable Populations	Targeted support programs for vulnerable populations, including refugees and low-income individuals. Community-based interventions and culturally sensitive approaches to engage vulnerable populations. Collaboration with non-governmental organizations and community organizations to deliver tailored support services and resources.	Policymakers, non-governmental organizations, community organizations
Governmental and Financial Aspects	Prioritize cancer control initiatives in national healthcare agendas and allocate sufficient resources. Reform health insurance policies to ensure comprehensive coverage for cancer care services. Public–private sector partnerships to leverage resources and expertise for cancer control efforts.	Ministry of Health, government agencies, insurance agencies

## Data Availability

The data presented in this study are available on request from the corresponding authors. The data are not publicly available due to privacy.

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
