# Peer review of "Systemic Barriers to Optimal Cancer Care in Resource-Limited Countries: Jordanian Healthcare as an Example"

_cancers, 2024, doi:10.3390/cancers16061117_

Round 1
Reviewer 1 Report (Previous Reviewer 1)
Comments and Suggestions for Authors
Revisions are well noted.
Author Response
The authors would like to thank the respected reviewer for their time and effort in critically appraising our scholarly work. Your comments were appreciated and were tended to appropriately and to the best of our ability. Thank you, again, for considering our review for publication.
Reviewer 2 Report (Previous Reviewer 3)
Comments and Suggestions for Authors
Dear authors,
Many thanks for resubmitting your work to the journal. My previous comments have been adequately addressed. The editor has the ultimate authority to make the final decision based on the journal's standards.
Best regards
Author Response
The authors would like to thank the respected reviewer for their time and effort in critically appraising our scholarly work. Your comments were appreciated and were tended to appropriately and to the best of our ability. Thank you, again, for considering our review for publication.
This manuscript is a resubmission of an earlier submission. The following is a list of the peer review reports and author responses from that submission.
Round 1
Reviewer 1 Report
Comments and Suggestions for Authors
Important topic and extensive work carried out.
Manuscript is not showing the materials, methods and data sources. After introduction it is jumping to findings. Add a section on methods and data- indicate which were the data sources used? Were they available in public domain? What is the validity of these sources? How was the review conducted? Which period was covered etc
Then results- now results also includes some commentary and a bit of discussion. Keep results as a straightforward set of findings from the review.
Discussion- discuss the strengths and limitations of the study and compare the findings with similar countries.
Potential solutions to the barriers can be exactly as per the table of barriers- with indication of who has the responsibility to do them.
Reviewer 2 Report
Comments and Suggestions for Authors
The paper is clear, it describes very well the pivotal elements causing the inadequacy of cancer care, focusing on the different kind of barriers. They involve both the social and the institutional characteristics which hinder the access to cancer care.
It could be interesting to describe an hypothetical sustainable cancer care system, which may be applied in this framework. It should take into account both the limits and the resources, considering the progresses already achieved.
Author Response
Firstly, we would like to thank the editorial office and the respected reviewers for their time and effort in critically appraising our scholarly work. Your comments are fully appreciated and were tended to appropriately and to the best of our ability.
Reviewer 2:
Comment: The paper is clear, it describes very well the pivotal elements causing the inadequacy of cancer care, focusing on the different kind of barriers. They involve both the social and the institutional characteristics which hinder the access to cancer care. It could be interesting to describe a hypothetical sustainable cancer care system, which may be applied in this framework. It should take into account both the limits and the resources, considering the progresses already achieved.
Response: We thank the esteemed reviewer for this important comment. We agree that envisioning a hypothetical sustainable cancer care system within the framework of our study could be immensely valuable. We incorporated a detailed description of such a system, considering the existing limitations and available resources. We additionally added a table that summarizes potential solutions to barriers in cancer care to resource limited countries, which is referred to table-2 in the manuscript. We included a section for each barrier, proposed solutions, and responsible entity. The discussion section now reads as follow:
“Drawing upon lessons learned from Jordan, a sustainable cancer care system in LMICs can be designed to address key challenges while maximizing available resources (Table 2). A comprehensive healthcare delivery model would integrate primary care services with specialized cancer centers to ensure seamless patient management and coordinated multidisciplinary care (Smith et al., 2018; King Hussein Cancer Center, 2021). Equitable access can be achieved by strategically allocating healthcare facilities and resources based on population needs, leveraging mobile clinics and telemedicine to bridge geographical gaps and improve access (Khatib et al., 2020; Lecadet et al., 2016).
Capacity building involves investing in training and retaining a skilled healthcare workforce through collaborative partnerships with international institutions and tailored educational programs (Hundt et al., 2012; Obeidat et al., 2017). Some LMICs have established cancer registries at federal level, however the quality control mechanisms of these registries are extremely variable, and quite often data is either missing, or lags current data entry mechanisms and long-term outcomes which leads to consider delays in updating the data and would preclude accurate mortality or morbidity analysis (Soliman et. al., 2019; Al-Shamsi et. al., , 2013; Bazarbashi et. al., 2017). The need of accurate data of cancer cases in a country is imperative to develop cancer control programs and screening guidelines. Unless the burden of each cancer type and subtype is known, it is hard to allocate appropriate resources for prevention or treatment of the specific types of cancer.
Financial support strategies include strengthening existing government initiatives and NGO partnerships to expand financial assistance programs, ensuring inclusivity in health insurance coverage, and providing targeted support for vulnerable populations (Khader et al., 2023; Marzouk et al., 2019). In addition, prioritizing prevention and early detection entails evidence-based public education campaigns and subsidized screening services (Abdulrahim & Jawad, 2018; Miles et al., 2004). Quality improvement initiatives would establish robust monitoring and benchmarking systems to track care quality across healthcare institutions, implementing standardized performance metrics and best practices informed by international standards and research evidence (Khoury & Mawajdeh, 2004; Hammad, 2016).
Medical training and research efforts would foster a culture of research infrastructure and collaborative partnerships between academia, healthcare institutions, and industry stakeholders (Al Dossary, 2018; Taha et al., 2012; Abdel-razeq et. al., 2020). Community engagement strategies would involve engaging local communities and grassroots organizations in cancer prevention and control efforts through culturally sensitive interventions and community-based participatory research approaches (Alawa et al., 2019; Ay, Arcos González, & Castro Delgado, 2016). Additionally, ensuring comprehensive palliative care services integrated within the cancer care continuum is crucial. This includes addressing physical, psychosocial, and spiritual needs of patients and their families, providing symptom management, pain relief, and end-of-life care (Shamieh et. al., 2023). By implementing evidence-based strategies within this framework, Jordan can overcome existing challenges and optimize its cancer care system to ensure equitable access to high-quality care for all residents, while acknowledging and addressing resource constraints and limitations.”
Please note that we had also changed the title of the manuscript to reflect upon this comparison, and to make our findings and solutions more generalizable to other resource- limited countries. The title of the manuscript now reads:
“Systemic Barriers to Optimal Cancer Care in Resource-Limited Countries: Jordanian Healthcare as an Example”.
Reviewer 3 Report
Comments and Suggestions for Authors
Dear Authors,
Many thanks for submitting your work to the journal. Upon reviewing your manuscript, many concerns were raised. Specifically:
The manuscript lacks international impact and interest due to the dissemination of data related to the Jordanian healthcare system.
The present study is just a narrative review without a systematic way of synthesizing and presenting the reviewed data.
The results section must be reorganized. There are a lot of comments and proposals that should be encompassed in the "Discussion" section.
Best regards
Author Response
Firstly, we would like to thank the editorial office and the respected reviewers for their time and effort in critically appraising our scholarly work. Your comments are fully appreciated and were tended to appropriately and to the best of our ability.
Reviewer 3:
Overall: Dear Authors, Many thanks for submitting your work to the journal. Upon reviewing your manuscript, many concerns were raised.
Comment: The manuscript lacks international impact and interest due to the dissemination of data related to the Jordanian healthcare system.
Response: We thank the esteemed reviewer for this important comment. We . We agree that drawing conclusions from our results to generalize to other resource-limited countries is immensely valuable. To enhance generalizability/ dissemination of data; we added the following:
- a) We added a section that compares of our findings in Jordan with similar resource limited and low-middle income countries. The section reads as follow:
“As the first of its kind focused on Jordan, the findings of our review serve as a foundation for addressing gaps in cancer care within the country. Moreover, the study’s insights are highly applicable to LMICs facing similar healthcare challenges and can be extrapolated to the Middle East and North Africa (MENA) region. While each country may have its unique challenges, several common themes emerge across the region. In the MENA region, limited health insurance coverage poses a significant barrier to accessing cancer care. Studies have shown that a lack of comprehensive health insurance schemes results in high out-of-pocket expenses for cancer treatment, leading to financial burdens that deter patients from seeking timely care (Sibai et al., 2014; Awad et al., 2019). This problem is exacerbated by the absence of government-funded healthcare programs tailored to cancer prevention, screening, diagnosis, and treatment (Knaul et al., 2015). For example, in Egypt, where health insurance coverage is limited and out-of-pocket payments are common, access to essential cancer services remains a challenge for many individuals, particularly those from low-income backgrounds (Kamal et al., 2021).
The inadequacy of healthcare infrastructure is another critical barrier to cancer care In LMICs. Limited access to oncology centers, diagnostic facilities, and essential medical equipment hinders early detection and timely treatment initiation (Nugent et al., 2021). In Iraq, the healthcare infrastructure has been severely affected by years of conflict and underinvestment, resulting in limited access to specialized cancer care services, particularly in rural areas (Al-Hadithi et al., 2020). The shortage of trained healthcare professionals, including oncologists, nurses, and allied healthcare workers, poses a significant challenge to cancer care provision in LMICs. In countries like Syria and Yemen, ongoing conflicts and political instability have disrupted medical education and training programs, leading to a severe scarcity of qualified personnel capable of delivering comprehensive cancer care services (El-Jardali et al., 2019; Al-Nsour et al., 2020). Disparities in access to cancer care between urban and rural areas are pervasive in LMICs. In Lebanon, there is a significant disparity in cancer care access between urban and rural areas, with rural populations facing challenges related to limited healthcare infrastructure and transportation barriers (Saleh et al., 2018). Similarly, in Saudi Arabia, efforts to improve cancer care access in rural areas have been hampered by logistical challenges and disparities in healthcare resource allocation (Jazieh et al., 2020).
Governmental policies and regulatory frameworks play a crucial role in shaping cancer care delivery in LMICs. However, weak governance, corruption, and a lack of political will to prioritize cancer control initiatives hinder progress in this area (El Saghir et al., 2020). Jordan has implemented national cancer control strategies and policies to improve cancer care delivery and outcomes, despite facing challenges related to limited resources and refugee healthcare (Al-Mousa et al., 2017). The provision of cancer care to refugees presents unique challenges in LMICs hosting large refugee populations. In Turkey, which hosts millions of Syrian refugees, efforts to provide cancer care to refugees have been hindered by language barriers, legal restrictions, and limited access to healthcare services in refugee camps (Sencan et al., 2020). Similarly, in Lebanon, where a large refugee population resides, integrating refugee healthcare into national cancer control programs has been challenging due to resource constraints and legal barriers (Spiegel et al., 2019).”
- b) Incorporated a detailed description of such a system, considering the existing limitations and available resources. We additionally added a table that summarizes potential solutions to barriers in cancer care to resource limited countries, which is referred to table-2 in the manuscript. We included a section for each barrier, proposed solutions, and responsible entity. The discussion section now reads as follow:
“Drawing upon lessons learned from Jordan, a sustainable cancer care system in LMICs can be designed to address key challenges while maximizing available resources (Table 2). A comprehensive healthcare delivery model would integrate primary care services with specialized cancer centers to ensure seamless patient management and coordinated multidisciplinary care (Smith et al., 2018; King Hussein Cancer Center, 2021). Equitable access can be achieved by strategically allocating healthcare facilities and resources based on population needs, leveraging mobile clinics and telemedicine to bridge geographical gaps and improve access (Khatib et al., 2020; Lecadet et al., 2016).
Capacity building involves investing in training and retaining a skilled healthcare workforce through collaborative partnerships with international institutions and tailored educational programs (Hundt et al., 2012; Obeidat et al., 2017). Some LMICs have established cancer registries at federal level, however the quality control mechanisms of these registries are extremely variable, and quite often data is either missing, or lags current data entry mechanisms and long-term outcomes which leads to consider delays in updating the data and would preclude accurate mortality or morbidity analysis (Soliman et. al., 2019; Al-Shamsi et. al., , 2013; Bazarbashi et. al., 2017). The need of accurate data of cancer cases in a country is imperative to develop cancer control programs and screening guidelines. Unless the burden of each cancer type and subtype is known, it is hard to allocate appropriate resources for prevention or treatment of the specific types of cancer.
Financial support strategies include strengthening existing government initiatives and NGO partnerships to expand financial assistance programs, ensuring inclusivity in health insurance coverage, and providing targeted support for vulnerable populations (Khader et al., 2023; Marzouk et al., 2019). In addition, prioritizing prevention and early detection entails evidence-based public education campaigns and subsidized screening services (Abdulrahim & Jawad, 2018; Miles et al., 2004). Quality improvement initiatives would establish robust monitoring and benchmarking systems to track care quality across healthcare institutions, implementing standardized performance metrics and best practices informed by international standards and research evidence (Khoury & Mawajdeh, 2004; Hammad, 2016).
Medical training and research efforts would foster a culture of research infrastructure and collaborative partnerships between academia, healthcare institutions, and industry stakeholders (Al Dossary, 2018; Taha et al., 2012; Abdel-razeq et. al., 2020). Community engagement strategies would involve engaging local communities and grassroots organizations in cancer prevention and control efforts through culturally sensitive interventions and community-based participatory research approaches (Alawa et al., 2019; Ay, Arcos González, & Castro Delgado, 2016). Additionally, ensuring comprehensive palliative care services integrated within the cancer care continuum is crucial. This includes addressing physical, psychosocial, and spiritual needs of patients and their families, providing symptom management, pain relief, and end-of-life care (Shamieh et. al., 2023). By implementing evidence-based strategies within this framework, Jordan can overcome existing challenges and optimize its cancer care system to ensure equitable access to high-quality care for all residents, while acknowledging and addressing resource constraints and limitations.”
- c) Please note that we had also changed the title of the manuscript to reflect upon this comparison, and to make our findings and solutions more generalizable to other resource- limited countries. The title of the manuscript now reads:
“Systemic Barriers to Optimal Cancer Care in Resource-Limited Countries: Jordanian Healthcare as an Example”.
Comment: The present study is just a narrative review without a systematic way of synthesizing and presenting the reviewed data.
Response: We thank the esteemed reviewer for this important comment. We have critically evaluated our approach and made significant revisions to provide a structured, comprehensive overview of the review conduct, data sources, eligibility criteria, and review methodology employed in our study. We have also requested the search strategy from the medical librarian who performed the search, which we attached with the manuscript as Supplemental File A.
In the revised manuscript, the Materials and Methods section now reads as follow:
“Materials and Methods
Data Sources and Validity
The data for this narrative review were gathered from a comprehensive search of peer-reviewed academic literature, government reports, and international organization databases. PubMed, EMBASE, Scopus, Web of Science were systematically searched using relevant keywords related to cancer care barriers in Jordan. Furthermore, the records of Google Scholar served as a supplementary resource to ensure a more comprehensive exploration of relevant studies. The search was manually conducted with the assistance of a certified medical librarian, and detailed search strategy for databases can be found in supplementary file A. Additionally, non-peer-review reports from organizations such as the World Health Organization (WHO), Jordanian Ministry of Health, KHCC reports, Jordanian Cancer Registry, and international NGOs working in Jordan were consulted. All data sources used in this review were available in the public domain and were deemed valid based on peer-review processes, government oversight, and institutional credibility. All processes, including systematic search conduction and documents, were conducted per the guidelines of the Preferred Reporting Items for Systematic Reviews and Meta-Analyses (PRIMSA) statement (Page et. al., 2021).
Review Conduct
The review was conducted in accordance with established narrative review methodologies. Initially, relevant articles and reports were identified through keyword searches and title screening. Abstracts were then reviewed to assess eligibility based on relevance to the topic of cancer care barriers in Jordan. Full-text articles and reports meeting inclusion criteria were thoroughly analyzed to extract relevant information on barriers to cancer care, including medical services and infrastructure, human resources, quality management, public awareness, socioeconomic disparities, and the specific challenges faced by vulnerable populations, with a focus on refugees. Data extraction included information on the nature of barriers, contributing factors, and proposed solutions. The review covered a period spanning from inception to January 3rd, 2023, providing a comprehensive analysis of the barriers to cancer care in Jordan over the past two decades.
Eligibility criteria
This review included studies that met the following criteria: studies that directly addressed topics related to cancer care, encompassing oncology services, healthcare infrastructure, medical services, healthcare facilities, oncology professionals, healthcare workforce, healthcare quality, quality management, cancer awareness, public education programs, health insurance, health disparities, socioeconomic status, refugee healthcare, government initiatives, financial assistance programs, healthcare policies, barriers to cancer care, and the Jordanian healthcare system. All study types including observational studies, reviews, opinion pieces, editorials, and conference abstracts were included in the search. The authors excluded studies/papers that were not conducted in Jordan. Studies not in the English language were also excluded to ensure accessibility and comprehension by the reviewers.
Analysis Approach
Data analysis involved synthesizing information from diverse sources to identify common themes, patterns, and key insights related to barriers to cancer care in Jordan. Thematic analysis techniques were employed to categorize and organize findings according to thematic areas, including healthcare infrastructure, human resources, quality management, public awareness, socioeconomic disparities, and refugee healthcare. The review also examined proposed policy measures and initiatives aimed at overcoming identified barriers, highlighting successful strategies and areas requiring further attention.”
Comment: The results section must be reorganized. There are a lot of comments and proposals that should be encompassed in the "Discussion" section.
Response: We thank the esteemed reviewer for this important comment. We have streamlined the Results section to present findings without commentary or discussion, ensuring a straightforward presentation of review outcomes. We have moved any information that does not contain results to the discussion section. Furthermore, we moved data that contains pertinent results from the discussion to the results section. These changes have been tracked throughout the manuscript and can be found in the results and discussion sections.